# Successful Hematopoietic Stem Cell Transplantation from a Matched Related Donor with Beta-Thalassemia Minor for Severe Aplastic Anemia

**DOI:** 10.3390/children7100162

**Published:** 2020-10-04

**Authors:** Mi Young Jung, Young Tae Lim, Hyunji Lim, Jeong Ok Hah, Jae Min Lee

**Affiliations:** 1Department of Pediatrics, Yeungnam University College of Medicine, Daegu 42415, Korea; jhjh3216@gmail.com (M.Y.J.); blueray14@nate.com (Y.T.L.); 21220028@ynu.ac.kr (H.L.); 2Department of Pediatrics, Daegu Fatima Hospital, Daegu 41199, Korea; johah@med.yu.ac.kr

**Keywords:** aplastic anemia, beta-thalassemia, thalassemia minor, hematopoietic stem cell transplantation

## Abstract

The first-line treatment for severe aplastic anemia (SAA) patients is hematopoietic stem cell transplantation (HSCT), with full-matched related donors considered the most suitable. We report a case of SAA in which the patient successfully underwent HSCT from a donor with β-thalassemia minor. The patient in this case underwent HSCT from a human leukocyte antigen (HLA)-matched younger brother with β-thalassemia minor. A 7-year-old girl was referred to our facility following a 6-month history of easy bruising and pallor. Laboratory examinations showed pancytopenia and hypocellular bone marrow with cellularity of <5%. She was diagnosed with acquired SAA, and HLA typing of her family members was performed. Her younger brother was an HLA-matched sibling but had β-thalassemia minor. Since his hemoglobin levels were maintained at 10–11 d/dL, he was considered a suitable HSCT donor. The conditioning regimen included fludarabine, cyclophosphamide, and anti-thymocyte globulin. The CD34+ and CD3+ cell counts were 6.6 × 10^6^/kg and 0.48 × 10^8^/kg, respectively. White blood cell engraftment was evident on day +11. Regimen-associated toxicities, such as anorexia and enteritis, were mild; no infections occurred, and no symptoms of acute graft-versus-host disease (GVHD) were observed. The 30-day follow-up bone marrow examination revealed normocellular marrow with 80%–90% cellularity. Acute or chronic GVHD has not been reported, and good performance status has been observed throughout the 5 years after HSCT. β-thalassemia minor patients can be considered as bone marrow donors for SAA patients.

## 1. Introduction

Aplastic anemia is a rare disorder characterized by pancytopenia and a hypocellular bone marrow [1]. The pathogenesis of aplastic anemia includes the number of hemopoietic progenitor cells, an increased number of suppressor T-cells releasing IFN-γ, and abnormalities in mesenchymal stem cells with a reduced ability to inhibit T-cell function [2].

According to the treatment guidelines from the Bone Marrow Committee of the Korean Society of Pediatric Hematology-Oncology, the treatment of choice for severe aplastic anemia (SAA) in children is hematopoietic stem cell transplantation (HSCT) from a human leukocyte antigen-matched family donor (HLA-MFD). For children without an MFD, HSCT from a matched unrelated donor (MUD) or immunosuppressive therapy (IST) in combination with anti-thymocyte globulin (ATG) and cyclosporine (CSA) has been a therapeutic option [3]. Alternative donor transplantations for patients non-responsive to IST include mismatched unrelated donor HSCT, umbilical cord blood HSCT, and haploidentical HSCT [4]. Until now, there have been no reports on HSCT from an MFD with β-thalassemia as an alternative donor following IST failure. We hereby report a case of successful HSCT from a hematologically stable MFD with β-thalassemia as an alternative donor in a patient without an MFD who had failed to respond to IST.

## 2. Case Report

A 7-year-old girl was referred to Yeungnam University Hospital following a 6-month history of easy bruising and pallor. Physical examination revealed multiple ecchymoses on her extremities. The patient was the first child of a non-consanguineous, native Korean couple. She was pale but had no palpable lymphadenopathy or hepatosplenomegaly. Her neurological examination findings were normal. The initial complete blood count revealed a leukocyte count of 3.1 × 10^9^/L, hemoglobin (Hb) level of 7.3 g/dL, platelet count of 16 × 10^9^/L, absolute neutrophil count of 0.899 × 10^9^/L, and corrected reticulocyte count of 0.59%. The initial bone marrow biopsy showed a hypocellular marrow with 40% cellularity that did not fit the diagnosis of SAA. Other laboratory findings were: C-reactive protein level, 0.092 mg/dL (normal range, <0.5 mg/dL), erythrocyte sedimentation rate, 42 mm/h (normal range, 0–20 mm/h); C3 level, 89 mg/dL (normal range, 83–177 mg/dL); C4 level, 23 mg/dL (normal range, 15–45 mg/dL); direct and indirect Coombs test, negative; antinuclear antibody, negative; blood cytomegalovirus polymerase chain reaction test, negative; anti-Epstein–Barr virus viral capsid antigen immunoglobulin M (IgM), negative; anti-hepatitis A virus IgM, negative; anti-hepatitis C virus antibody, negative; hepatitis B surface antigen, negative; and anti-mycoplasma IgM, positive. Despite positivity for anti-mycoplasma IgM, she had no respiratory symptoms or any signs of infection; therefore, antibiotic administration was not indicated. She had no dysmorphic features and no cutaneous pigmentation, nail dystrophy, leukoplakia, or skeletal anomaly was found. Inherited bone marrow failure syndrome was not suspected, and the chromosomal breakage test showed normal results. The telomere length test was not available at that time. A diagnosis of non-SAA was assigned to the patient, and treatment was not indicated at this stage.

A repeat bone marrow examination was performed 10 months later because of persistent pancytopenia. Her bone marrow was hypocellular with less than 5% cellularity, and complete blood count showed a white blood cell (WBC) count of 3.43 × 10^9^/L, Hb level of 7.7 g/dL, platelet count of 13 × 10^9^/L, absolute neutrophil count of 0.871 × 10^9^/L, and corrected reticulocyte count of 0.76%. Therefore, she was diagnosed with SAA at that time.

We considered HSCT for this patient and performed HLA typing and blood tests of her family members to find a suitable donor. Her younger brother was an HLA-matched sibling but had mild anemia with Hb levels maintained at approximately 10–11 g/dL. The whole-exome sequencing test revealed a heterozygous frameshift variant of Hemoglobin Subunit Beta (c.27dupG, p.Ser10Valfs*14), a mutation that has previously been reported in patients with β-thalassemia (Figure 1) [5].

No HLA-identical matched related donors other than her brother were identified. HSCT from an HLA-identical donor with β-thalassemia minor has raised skepticism regarding its therapeutic outcome. Therefore, we opted for IST, which was performed with ATG, CSA, and prednisolone. CSA was administered for 15 months with a partial response. However, severe cytopenia was observed 15 months later that required red blood cell (RBC) and platelet transfusions.

After confirming that there was no response to IST, we searched for unrelated matched donors, but no suitable unrelated full match donor or an unrelated mismatch donor was found. In addition, there were concerns about HSCT from the parents who were mismatched donors. Hence, we could not find possible alternative donors such as haplo-identical family donors.

Since her brother who was 7 years old at the time demonstrated hematological stability, with an Hb level maintained at 10–11 g/dL, we considered HSCT to be feasible as a second-line treatment. The conditioning regimen included intravenous fludarabine (30 mg/m^2^/day from day −5 to day −2), cyclophosphamide (25 mg/kg/day from day −5 to day −2), and thymoglobulin (2.5 mg/kg/day from day −3 to day −1). To prevent graft-versus-host disease (GVHD), CSA (3 mg/kg/day from day −1) was administered with drug level monitoring. Additionally, intravenous methotrexate was administered at a dose of 5 mg/m^2^/day on days +1, +3, +6, and +11. Hematopoietic stem cells were harvested from her brother’s peripheral blood through a femoral catheter after four days of G-CSF administration to the day of transplantation, according to the allogeneic PBSCT protocol.

The total nucleated cell, CD34+, and CD3+ cell counts were 2.35 × 10^8^/kg, 6.6 × 10^6^/kg, and 0.48 × 10^8^/kg, respectively. WBC engraftment was performed on day +11 (The day of engraftment was defined as the first of three consecutive days on which the granulocyte count exceeded 0.5 × 10^9^/L).

Regimen-associated toxicities, such as anorexia and enteritis, were mild. No infection occurred, and there were no symptoms of acute GVHD. The 30-day follow-up bone marrow examination revealed normocellular marrow with a cellularity of 80–90%, and chimerism analysis by variable-number tandem repeat revealed full donor chimerism (>95% donor type). The chimerism analysis on day 114 also revealed full donor chimerism. Acute or chronic GVHD was not observed. Furthermore, good performance status has been observed for 5 years after HSCT. The patient’s complete blood count showed normal results and no further blood transfusions have been required to this day; her younger brother also did not require transfusion.

## 3. Discussion

SAA is characterized by pancytopenia, a reduction in the counts of all blood cells (RBCs, WBCs, and platelets) and the absence of infiltration and fibrosis in the bone marrow [6]. According to the Bone Marrow Committee of the Korean Society of Pediatric Hematology-Oncology treatment guideline, the treatment of choice for SAA in children is HSCT from an HLA-MFD. For children lacking an MFD, IST with a combination of ATG and CSA has been used as a therapeutic option [3].

In our case, the patient’s younger brother was an MFD but had β-thalassemia minor. Due to the lack of other suitable MUDs, compounded by the lack of clinical evidence on transplant outcomes from an MFD with β-thalassemia, we initially decided to administer IST.

The SAA Working Party of the European Society of Blood and Marrow Transplantation announced that the 10-year overall survival (OS) rate of children treated with first-line IST had improved from 81% (*n* = 304 from 1991 to 2002) [7] to 87% (*n* = 167 from 2000 to 2009) [8]. Furthermore, a Japanese study conducted between 1992 and 2009 (*n* = 386) reported a 10-year OS rate of 88% in patients with SAA treated with first-line IST [9]. The OS rate reported with IST was comparable to those reported with MFD HSCT. In contrast to the high OS rate, event-free survival after IST was reported as far less satisfactory than that following MFD HSCT (33% vs. 87%, *p* = 0.001) [8].

After 15 months of IST, no clinical improvement was observed, and no alternative MUD was found. Despite the β-thalassemia minor phenotype, her brother was sufficiently hematologically stable to maintain an Hb level of 10–11 g/L. After careful consideration, we opted for HSCT despite the lack of clinical evidence.

β-thalassemia belongs to a group of hereditary blood disorders characterized by reduced or absent β-globin chain synthesis, resulting in variable phenotypes ranging from severe anemia to a clinically asymptomatic state. Anemia has two etiologies: ineffective erythropoiesis and hemolysis. The underlying mechanism is thought to be related to an imbalance between alpha and beta chain production, which tend to aggregate and precipitate and in turn, damage the RBC membrane. The RBCs then become dehydrated and/or rigid, making them less deformable as they move through the vasculature and the reticuloendothelial system [10]. Both anemia and increased erythropoiesis lead to suppression of hepcidin and, therefore, increased iron absorption [11].

According to the suitability criteria for adult related donors, hematopoietic stem cell collections from individuals with RBC abnormalities such as spherocytosis and elliptocytosis are generally not recommended; however, subjects with glucose-6-phosphate dehydrogenase deficiency and thalassemia traits, mild α-thalassemia, or β-thalassemia minor are suitable hematopoietic stem cell donors [12]. Nevertheless, to our best knowledge, no cases of pediatric SAA patients undergoing HSCT from donors with β-thalassemia have been reported yet. β-thalassemia minor is usually asymptomatic but sometimes involves mild anemia, which does not require treatment [13]. However, considering the pathophysiology of β-thalassemia, our main concern was the risk of systemic complications such as hepatic, endocrine, and cardiac dysfunctions after HSCT from an MFD with β- thalassemia [14].

In HSCT for SAA, bone marrow as a stem cell source is recommended because it causes less GVHD and yields better outcomes [15]. Recently, survival outcomes were seen to improve with G- CSF-mobilized allogeneic peripheral blood stem cells using fludarabine base conditioning in children and adults [16,17]. Regarding the haplo-identical family donor (HFD), although Kim et al. reported excellent transplant outcomes from halo-identical donors using selective ex-vivo T cell depletion, it is not yet a commonly used method due to the high cost in Korea. Further, GVHD and other complications such as cytomegalovirus or Epstein–Barr virus infection, and cystitis were still more prevalent in HFD HSCT [18].

In our case, the conditioning regimen was composed of cyclophosphamide, fludarabine, and ATG, followed by GVHD prophylaxis with methotrexate and CSA. Preconditioning with cyclophosphamide, fludarabine, and ATG was superior to that with cyclophosphamide-ATG in terms of fewer regimen-related toxicities without an increase in the rate of engraftment failure (23.3% vs. 55.0%, *P* = 0.003) [16,17]. WBC engraftment and platelet engraftment were performed on day +11 and day +24, respectively. Treatment-related toxicities, such as GVHD, veno-occlusive disease, and thrombotic microangiopathy, were not observed and no infections occurred.

Our case findings suggest that HLA-matched siblings with β-thalassemia minor could be suitable HSCT donors for SAA patients.

## Figures and Tables

**Figure 1 children-07-00162-f001:**
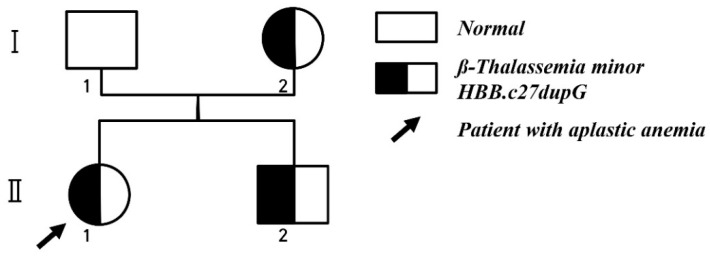
Pedigree of proband (black arrowhead) with aplastic anemia and β-thalassemia minor.

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
