# Peer review of "Successful Hematopoietic Stem Cell Transplantation from a Matched Related Donor with Beta-Thalassemia Minor for Severe Aplastic Anemia"

_children, 2020, doi:10.3390/children7100162_

Round 1

Reviewer 1 Report

General comments for the authors

Jung et al. report a SAA patient that received successful HSCT from an HLA identical brother with betathalassemia minor (thalassemia trait). Although family donors with betathalassemia minor are generally considered as suitable donors, descriptions of their use in pediatric patients are scarce. This is probably because HLA matched register donors are viable option for patients with betathalassemia major (Bernando et al. Blood 2012;120:473). I think that this case report should be revised and consider following suggestions and comments:

Major comments:

  1. Authors state that “inherited bone marrow failure syndrome was not suspected”. Chromosomal breakage test was normal, but otherwise authors do not present diagnostic work-up of inherited bone marrow syndromes. E.g. inherited defects in DNA repair or telomeropathies are important to exclude because patients with these diseases bear particular sensitivity to toxicities. Authors should present status findings (e.g. dysmorphic features and signs of dyskeratosis or their absence should be noted) and what kind of genetic analysis was done. If genetic analysis was not done this should be clearly indicated and reasons for omitting it should be presented (authors present the betathalassemia mutation of the donor).
  2. Why father was not considered as a haploidentical donor? He is the only family member without thalassemia trait and Korea group has published excellent results using a/b TCR depleted haploidentical grafts (Kim et al. Biol Blood Marrow Transplant 2019;25:965).
  3. Why PBSC:s were used? Bone marrow graft would be classical choice for patient with SAA as it bears lower risk of GVHD. Furthermore, many pediatric HSCT centers avoid G-CSF mobilization of under aged family donors. Choice of donor should be discussed in more detail.
  4. URD could not be found for the patient as the authors state in the Discussion. This important fact should be included in the more detailed description of donor selection process that should be provided already in Case Report section. Criteria for URD selection should also described (which mismatch grade could be accepted, etc.). The most important message in this case report is the donor selection process which ended into conclusion that MFD with thalassemia minor was the best option. The selection process should be described in detail and presented very clearly.
  5. WBC engraftment was evident on day +11 and platelet engraftment on day +24. How authors define engraftment? Definition of engraftment should be given in the manuscript as well as the day of neutrophil engraftment (e.g. the day when ANC > 0.5 x 109/L permanently). Details of engraftment should be given in the Case Report section and it is not necessary to repeat this information in Discussion.
  6. Did the authors measure chimerism from peripheral blood and/or bone marrow after the transplant? Results of chimerism analysis or reason of not measuring chimerism should be given in the manuscript.

Minor comments:

  1. Authors repeatedly write that “engraftment was performed” (in Abstract, Case Report, and Discussion). Better expression would be e.g. “WBC engraftment was evident on day +x”.
  2. On page 2 line 63 authors write “Therefore patient was diagnosed with non-SAA without any treatment”. Better formulation would be e.g. “Diagnosis of non-SAA was assigned to the patient while treatment was not indicated at this stage.”
  3. On page 3 line 120 authors state that RBC membrane damage in betathalassemia is due to excess production of alpha chains. I think that problem is better described as an imbalance between alpha and beta chain production.
  4. Short description of pathogenesis of SAA should be added to Introduction.

Author Response

Response to Reviewer 1 Comments

Major comments:

  1. Authors state that “inherited bone marrow failure syndrome was not suspected”. Chromosomal breakage test was normal, but otherwise authors do not present diagnostic work-up of inherited bone marrow syndromes. E.g. inherited defects in DNA repair or telomeropathies are important to exclude because patients with these diseases bear particular sensitivity to toxicities. Authors should present status findings (e.g. dysmorphic features and signs of dyskeratosis or their absence should be noted) and what kind of genetic analysis was done. If genetic analysis was not done this should be clearly indicated and reasons for omitting it should be presented (authors present the beta thalassemia mutation of the donor).

Thank you for your comment.

A supplementary explanation for inherited bone marrow failure syndrome and reasons for not performing genetic analysis were added.

  1. Why father was not considered as a haploidentical donor? He is the only family member without thalassemia trait and Korea group has published excellent results using a/b TCR depleted haploidentical grafts (Kim et al. Biol Blood Marrow Transplant 2019;25:965).

Regarding the haplo-identical family donor (HFD), although Kim et al. reported excellent transplant outcomes from halo-identical donors using selective ex-vivo T cell depletion, it is not yet a commonly used method due to the high cost in Korea. Further, GVHD and other complications such as cytomegalovirus or Epstein–Barr virus infection, and cystitis were still more prevalent in HFD HSCT [18]. (Lines 149-153)

  1. Why PBSC:s were used? Bone marrow graft would be classical choice for patient with SAA as it bears lower risk of GVHD. Furthermore, many pediatric HSCT centers avoid G-CSF mobilization of under aged family donors. Choice of donor should be discussed in more detail.

In HSCT for SAA, bone marrow as a stem cell source is recommended because it causes less GVHD and yields better outcomes [15]. Recently, survival outcomes were seen to improve with G-CSF-mobilized allogeneic peripheral blood stem cells using fludarabine base conditioning in children and adults [16, 17]. (Lines 146-149)

  1. URD could not be found for the patient as the authors state in the Discussion. This important fact should be included in the more detailed description of donor selection process that should be provided already in Case Report section. Criteria for URD selection should also described (which mismatch grade could be accepted, etc.). The most important message in this case report is the donor selection process which ended into conclusion that MFD with thalassemia minor was the best option. The selection process should be described in detail and presented very clearly.

After confirming that there was no response to IST, we searched for unrelated matched donors, but no suitable unrelated full match donor or an unrelated mismatch donor was found. In addition, there were concerns about HSCT from parents who were mismatched donors. Hence, we could not find possible alternative donors such as haplo-identical family donors.

 (Lines 84-87)

  1. WBC engraftment was evident on day +11 and platelet engraftment on day +24. How authors define engraftment? Definition of engraftment should be given in the manuscript as well as the day of neutrophil engraftment (e.g. the day when ANC > 0.5 x 109/L permanently). Details of engraftment should be given in the Case Report section and it is not necessary to repeat this information in Discussion.

A detailed definition of WBC engraftment was added to the case report section. (The day of engraftment was defined as the first of three consecutive days on which the granulocyte count exceeded 0.5×109/L) (Lines 98-99)

  1. Did the authors measure chimerism from peripheral blood and/or bone marrow after the transplant? Results of chimerism analysis or reason of not measuring chimerism should be given in the manuscript.

The 30-day follow-up bone marrow examination revealed normocellular marrow with a cellularity of 80%–90%, and chimerism analysis by variable number tandem repeat revealed full donor chimerism (>95% donor type). The chimerism analysis on day 114 also revealed full donor chimerism. (Lines 102-104)

Minor comments:

  1. Authors repeatedly write that “engraftment was performed” (in Abstract, Case Report, and Discussion). Better expression would be e.g. “WBC engraftment was evident on day +x”.

I have modified this according to your comment.

  1. On page 2 line 63 authors write “Therefore patient was diagnosed with non-SAA without any treatment”. Better formulation would be e.g. “Diagnosis of non-SAA was assigned to the patient while treatment was not indicated at this stage.”

I have modified this according to your comment.

  1. On page 3 line 120 authors state that RBC membrane damage in beta thalassemia is due to excess production of alpha chains. I think that problem is better described as an imbalance between alpha and beta chain production.

I have modified this according to your comment.

  1. Short description of pathogenesis of SAA should be added to Introduction.

The pathogenesis of aplastic anemia included the number of hemopoietic progenitor cells, an increased number of suppressor T-cells releasing IFN-γ, and abnormalities in mesenchymal stem cells with a reduced ability to inhibit T-cell function (Lines 32-34)

Reviewer 2 Report

Mi Y Jung et al.: Successful hematopoietic stem cell transplantation from a matched related donor with beta-thalassemia minor for severe aplastic anemia.

The authors are reporting a case presentation about the treatment course of a female patient of 7 years. She was presenting with clinical and laboratory findings of severe aplastic anemia. Treatment of choice is the allogeneic SCT from a matched related healthy donor (MFD). The only family who matched in his HLA genes to the patient was her “younger” brother who suffered from beta thalassemia minor without transfusion dependency.

As of concerns using a MFD with thalassemia minor, the authors started immune suppressive treatment over a period of 15 months without clinical improvement. Following this the decision was taken to use the young brother as donors.

Remarks:

This is an interesting observation and a valuable contribution to the medical experience.

I have two concerns:

  1. How old was the brother? If he was “younger” than the patient, the authors may discuss whether this donor still might become transfusion dependent in later life?
  1. As I understood from the paper, the authors used peripheral stem cells for transplantation. As G-CSF is not licensed for mobilization in minors for the donation of stem cells in the allogeneic setting, the authors should discuss why they did not opt for bone marrow as the standard treatment?
  2. Did the donor need a central line for apheresis?

Minor:

Although not a native speaker, I think, that the ms would benefit from minor English editing.

Author Response

Response to Reviewer 2 Comments

  1. How old was the brother? If he was “younger” than the patient, the authors may discuss whether this donor still might become transfusion dependent in later life?

Since her brother who was 7 years old at the time demonstrated hematological stability with a Hb level maintained at 10–11 g/dL, we considered HSCT to be feasible as a second-line treatment. (Lines 88-89)

  1. As I understood from the paper, the authors used peripheral stem cells for transplantation. As G-CSF is not licensed for mobilization in minors for the donation of stem cells in the allogeneic setting, the authors should discuss why they did not opt for bone marrow as the standard treatment?

In HSCT for SAA, bone marrow as a stem cell source is recommended because it causes less GVHD and yields better outcomes [15]. Recently, survival outcomes were seen to improve with G-CSF-mobilized allogeneic peripheral blood stem cells using fludarabine base conditioning in children and adults [16,17]. Regarding the haplo-identical family donor (HFD), although Kim et al. reported excellent transplant outcomes from halo-identical donors using selective ex-vivo T cell depletion, it is not yet a commonly used method due to the high cost in Korea. Further, GVHD and other complications such as cytomegalovirus or Epstein–Barr virus infection, and cystitis were still more prevalent in HFD HSCT [18]. (Lines 146-153)

  1. Did the donor need a central line for apheresis?

Peripheral blood stem cells were collected through a femoral catheter.

Hematopoietic stem cells were harvested from her brother’s peripheral blood through a femoral catheter after four days of G-CSF administration up to the day of transplantation, according to the allogeneic PBSCT protocol. (Lines 94-96)

Minor:

Although not a native speaker, I think, that the ms would benefit from minor English editing.

I submitted the manuscript for English proofreading.
